# Metabolomic Profile and Biological Properties of Sea Lavender (*Limonium algarvense* Erben) Plants Cultivated with Aquaculture Wastewaters: Implications for Its Use in Herbal Formulations and Food Additives

**DOI:** 10.3390/foods10123104

**Published:** 2021-12-14

**Authors:** Maria João Rodrigues, Viana Castañeda-Loaiza, Ivo Monteiro, José Pinela, Lillian Barros, Rui M. V. Abreu, Maria Conceição Oliveira, Catarina Reis, Florbela Soares, Pedro Pousão-Ferreira, Catarina G. Pereira, Luísa Custódio

**Affiliations:** 1Centre of Marine Sciences, Faculty of Sciences and Technology, Campus of Gambelas, University of Algarve, Ed. 7, 8005-139 Faro, Portugal; vcloaiza@ualg.pt (V.C.-L.); cagpereira@ualg.pt (C.G.P.); lcustodio@ualg.pt (L.C.); 2IPMA, Aquaculture Research Station, Av. do Parque Natural da Ria Formosa s/n, 8700-194 Olhao, Portugal; ivojmonteiro@gmail.com (I.M.); fsoares@ipma.pt (F.S.); pedro.pousao@ipma.pt (P.P.-F.); 3Centro de Investigação de Montanha (CIMO), Instituto Politécnico de Bragança, Campus de Santa Apolónia, 5300-253 Braganca, Portugal; jpinela@ipb.pt (J.P.); lillian@ipb.pt (L.B.); ruiabreu@ipb.pt (R.M.V.A.); 4Centro de Química Estrutural, Complexo Interdisciplinar, Instituto Superior Técnico, Universidade de Lisboa, Av. Rovisco Pais, 1049-001 Lisboa, Portugal; conceicao.oliveira@tecnico.ulisboa.pt; 5iMed.Ulisboa, Faculdade de Farmácia, Universidade de Lisboa, Av. Prof. Gama Pinto, 1649-003 Lisboa, Portugal; catarinareis@ff.ulisboa.pt

**Keywords:** halophytes, herbal products, IMTA systems, saline agriculture, salinization, salt tolerant plants, sustainability

## Abstract

Water extracts from sea lavender (*Limonium algarvense* Erben) plants cultivated in greenhouse conditions and irrigated with freshwater and saline aquaculture effluents were evaluated for metabolomics by liquid chromatography-tandem high-resolution mass spectrometry (LC-HRMS/MS), and functional properties by in vitro and ex vivo methods. In vitro antioxidant methods included radical scavenging of 2,2-diphenyl-1-picrylhydrazyl (DPPH) and 2,2′-azino-bis(3-ethylbenzothiazoline-6-sulfonic acid (ABTS), ferric-reducing antioxidant power (FRAP), and copper and iron chelating assets. Flowers’ extracts had the highest compounds’ diversity (flavonoids and its derivatives) and strongest in vitro antioxidant activity. These extracts were further tested for ex vivo antioxidant properties by oxidative haemolysis inhibition (OxHLIA), lipid peroxidation inhibition by thiobarbituric acid reactive substances (TBARS) formation, and anti-melanogenic, anti-tyrosinase, anti-inflammation, and cytotoxicity. Extract from plants irrigated with 300 mM NaCl was the most active towards TBARS (IC_50_ = 81 µg/mL) and tyrosinase (IC_50_ = 873 µg/mL). In OxHLIA, the activity was similar for fresh- and saltwater-irrigated plants (300 mM NaCl; IC_50_ = 136 and 140 µg/mL, respectively). Samples had no anti-inflammatory and anti-melanogenic abilities and were not toxic. Our results suggest that sea lavender cultivated under saline conditions could provide a flavonoid-rich water extract with antioxidant and anti-tyrosinase properties with potential use as a food preservative or as a functional ingredient in herbal supplements.

## 1. Introduction

The use of herbal medicines and supplements is widespread among patients with chronic health ailments [1], and as a result, the commercial interest in natural health supplements for improving biological functions and wellness is raising the market value of herbal supplements. The current market size value is USD 6.3 billion, and with a compound annual growth rate (CAGR) of 6.2% it is estimated to reach USD 8.5 billion by 2025. However, the European market is expected to record a CAGR of 8.2% [2,3]. Moreover, several studies show that the excessive consumption of synthetic food additives is related to several health problems, including attention deficit disorders, gut diseases, obesity, immune system problems, and allergies [4,5,6,7,8]. Consequently, an increasing number of consumers is doubting the use of artificial food additives, as they are increasingly being considered unhealthy [9], urging the need to find more efficient and safer alternatives and boosting a growing demand in the food industry for natural ingredients to create “clean label” products [10]. Different organisms have been explored as sources of bioactive ingredients to be used as herbal supplements and/or food additives, such as mushrooms, glycophytes, and algae [11,12]. However, despite the recognized biotechnological potential of salt tolerant plants, that is, halophytes [13], such plants remain mostly unexplored [14]. 

Halophytes have evolved to live in abiotic stressful environments (e.g., hypersalinity, high UV radiation, and fluctuating extreme temperatures) that enhance the production of free radicals and the occurrence of oxidative stress in plant cell tissues by developing diverse anatomical (e.g., alt glands, succulence) and biochemical (e.g., synthesis of specific solutes, antioxidant enzymes, and secondary metabolites, selective accumulation, or exclusion of specific ions) adaptations [15]. Specifically, secondary antioxidant metabolites which are produced, such as phenolic acids, flavonoids, sterols, and vitamins, are not only crucial for plant survival, but also exhibit beneficial properties for humans, such as antioxidant, anti-inflammatory, and antitumor properties, and are therefore of high interest for the food and pharmaceutical industries [16]. Halophytes are thus considered an important reservoir of compounds with biotechnological applications, and several species are currently being produced and commercially exploited as food, food supplements, and cosmetic ingredients, including quinoa (*Chenopodium quinoa* Willd.), sea asparagus (*Salicornia* spp.), sea fennel (*Chritmum maritimum* L.), and golden samphire (*Inula chritmoides* L.) [17,18,19]. Moreover, due to the high demand for more efficient and safer alternatives, the search for new sources of natural additives to replace the synthetic ones is rising in the food industry. For example, antioxidants and anti-browning agents are valuable for inhibiting detrimental changes of foods which are sensitive to oxidation. Due to halophytes’ chemical and biological richness, there is a growing interest in identifying halophyte species with potential uses in this sector that is thirsty for innovation [20]. 

Besides their commercial potential, halophytes have the advantage of being able to be cultivated in saline conditions, which is of particular importance in the context of the increasing scarcity of freshwater for agriculture. In fact, the United Nations (UN) have established several farming alternatives for sustainable crop production, which include the use of brackish and saltwater for irrigation. Salinity negatively affects the productivity of most of the conventional glycophytic crops, and therefore, halophytes are the most promising model for biosaline agriculture systems [21,22]. One of such systems is integrated multi-trophic aquaculture (IMTA), where aquaculture wastes are used as both irrigation and fertilization for halophytes [23,24]. IMTA systems contribute to the reduction of the environmental impacts of aquaculture while adding further income by producing crops with commercial food and/or medicinal uses, and were already optimized for the cultivation of different halophytic species, such as sea asparagus (*Salicornia* sp.) and sea purslane (*Halimiones portulacoides* (L.) Aellen) [23,24].

The halophyte *Limonium algarvense* Erben (sea lavender) is rich in several bioactive secondary metabolites, especially flavonoids and its derivatives, and displays important bioactivities that are highly relevant for the improvement of human health [25,26,27]. Moreover, this species can be cultivated under greenhouse conditions and irrigated with aquaculture wastewater up to 300 mM NaCl [28]. This work aimed to further explore sea lavender cultivated under irrigation with aquaculture wastewater as a source of innovative and sustainable natural products to be used as food preservatives and/or as a functional ingredient for herbal supplements. For that purpose, sea lavender water extracts were profiled for chemical and functional properties, including in vitro (antioxidant activity, tyrosinase inhibition, anti-melanogenic and anti-inflammatory properties, cytotoxicity) and ex vivo (antioxidant) methods. 

## 2. Materials and Methods

### 2.1. Chemicals

Sigma-Aldrich (Lisbon, Portugal) provided the 1,1-diphenyl-2-picrylhydrazyl (DPPH), 2,2’-azino-bis(3-ethylbenzothiazoline-6-sulphonic acid) (ABTS), 2,2′-azobis(2-methylpropionamidine) dihydrochloride (AAPH), butylated hydroxytoluene (BHT), and the mouse melanoma B16 5A4 cells. Further chemicals and solvents were supplied by VWR International (Leuven, Belgium). Methanol, acetonitrile, water LC-MS optima grade, and formic acid LC-MS grade were supplied by Fisher Scientific (Hampton, VA, USA). 

### 2.2. Plant Material

Cultivated plants were derived from wild-collected seeds (June 2018) of sea lavender from “Ria de Alvor” (Algarve, Portugal; coordinates: 37°07′34.8″ N 8°35′54.9″ W) and grown under greenhouse conditions irrigated with freshwater (approx. 0 mM NaCl) and aquaculture wastewaters at two salinity concentrations, namely, 300 and 600 mM NaCl [28]. After 14 weeks of cultivation, plants were separated into flowers, peduncles, and leaves, dried for 3 days at 40 °C, and powdered and stored at −20 °C until needed. 

### 2.3. Preparation of the Extracts

Dried biomass was extracted with distilled water by an ultrasound-assisted procedure (1:40, *w*/*v*) for 30 min (ultrasonic bath USC-TH (VWR, Leuven, Belgium), capacity of 5.4 L, frequency of 45 kHz, supply of 230 V, a tub heater of 400 W, temperature control made by a LED display). Extracts were filtered (Whatman n° 4), evaporated under reduced pressure and temperature in a rotary evaporator, weighted, dissolved at a concentration of 10 mg/mL in distilled water, and stored at −20 °C.

### 2.4. Liquid Chromatography-Tandem High-Resolution Mass Spectrometry (LC-HRMS/MS) Analysis

The extracts were analyzed by Liquid Chromatography (UHPLC Elute) interfaced with a QqTOF Impact II mass spectrometer equipped with an ESI source, operating at a negative mode (Bruker Daltonics, Bremen, Germany). Chromatographic separation was carried out on a C18 reversed-phase Halo column 100 Å (150 mm × 2.1 mm, 2.7 μm particle size; Advanced Materials Technology, Wilmington, DE, USA), using a gradient elution of 0.1% formic acid in water (phase A) and acetonitrile (phase B). For details on LC-HRMS/MS settings, see Rodrigues et al. [28].

### 2.5. In Vitro Antioxidant Activity

#### 2.5.1. Radical Scavenging Activity (RSA) on DPPH^•^ and ABTS^+•^

Samples and positive control (BHT) were tested for RSA against the DPPH and ABTS radicals at concentrations ranging from 10 to 1000 µg/mL, as described previously [25]. Results were expressed as inhibition percentage in relation to the negative control (distilled water), and as half-maximal inhibitory concentration (IC_50_ values, µg/mL), when possible.

#### 2.5.2. Ferric Reducing Antioxidant Power (FRAP) 

Samples’ ability to reduce Fe^3+^ was assessed by the method described by Rodrigues et al. [25]. Absorbance was measured at 700 nm (Biochrom EZ Read 400, Santa Clara, CA, USA), and increased absorbance of the reaction means higher reducing power. Results were expressed as a percentage relative to the standard (BHT, 1 mg/mL), and as IC_50_ values (µg/mL), when possible.

#### 2.5.3. Metal Chelating Activity on Iron (ICA) and Copper (CCA)

ICA and CCA were tested on samples and the standard (EDTA) at different concentrations (10–1000 µg/mL), as described earlier [25]. Differences in absorbance were assessed on a microplate reader (Biochrom EZ Read 400). Results were expressed as an inhibition percentage compared to the negative control (distilled water), and as IC_50_ values (µg/mL), whenever possible.

### 2.6. Ex Vivo Antioxidant Activity

#### 2.6.1. Oxidative Haemolysis Inhibition Assay (OxHLIA)

A sheep erythrocyte solution (2.8 %, *v*/*v*; 200 µL) prepared in phosphate-buffered saline (PBS, pH 7.4) was mixed with 400 µL of either: sample (0.0625–2 mg/mL), PBS (control), distilled water (baseline), or trolox (7.81–250 µg/mL). After pre-incubation at 37 °C for 10 min with shaking, 200 μL of AAPH (160 mM) were added and the optical density was measured kinetically at 690 nm (microplate reader Bio-Tek Instruments, ELX800, Santa Clara, CA, USA) until complete haemolysis [29]. Results were expressed as IC_50_ values (µg/mL) for a Δt of 60 min.

#### 2.6.2. Inhibition of Lipid Peroxidation by the Thiobarbituric Acid Reactive Substances (TBARS) Assay

A porcine brain cell solution (1:2, *w*/*v*; 100 µL) was incubated with 200 µL of sample (0.0625–2 mg/mL) or trolox (3.125–100 µg/mL) plus 100 µL of FeSO_4_ (10 µM) and 100 µL of ascorbic acid (0.1 mM) at 37 °C for 1 h. Then, 500 µL of trichloroacetic acid (28 % *w*/*v*) and 380 µL of thiobarbituric acid (TBA; 2 % *w*/*v*) were added and the mixture was heated at 80 °C for 20 min. After centrifugation, the color intensity of the malondialdehyde (MDA)-TBA complexes formed in the system was measured at 532 nm [30]. Results were expressed as IC_50_ values (µg/mL).

### 2.7. In Vitro Tyrosinase Inhibition

Samples were tested at concentrations between 10 and 1000 μg/mL for tyrosinase inhibitory activity, as previously described [31]. Results were expressed as an inhibition percentage relative to a control containing ultrapure water, and when possible as IC_50_ values (μg/mL). 

### 2.8. Cell Culture

The RAW 264.7 cells were maintained in RPMI 1640 culture media, whereas HEK 293, HepG2, and B16 4A5 cell lines were cultured in DMEM media. All media were supplemented with 10 % heat-inactivated fetal bovine serum, glutamine (2 mM), penicillin (100 U/mL), and streptomycin (100 mg/mL). All cell lines were kept in an incubator at 37 °C, with 5 % CO_2_ and under a humid atmosphere.

#### 2.8.1. In Vitro Anti-Inflammatory Activity

RAW 264.7 macrophages were seeded with a cell density of 5 × 10^5^ cells/mL in a 96-well microplate (300 µL/well) and left to adhere for 24 h. Afterwards, cells were treated with different concentrations of the samples (15 μL) at several concentrations (6.25–100 µg/mL) and incubated for 1 h. Then, 30 μL of liposaccharide (LPS; 1 mL/mL) was added to each well and incubated for an additional 24 h. Dexamethasone was used as a positive control. Quantification of nitric oxide was performed using a Griess reagent system kit (nitrophenamide, ethylenediamine, and nitrite solutions) using a sodium nitrite calibration curve. The samples’ absorbance was measured at 540 nm on a microplate reader (ELX800 Biotek, Santa Clara, CA, USA) and results were expressed as a percentage of inhibition of nitric oxide production relative to the negative control (ultrapure water), and, if possible, as half-maximal inhibitory concentration (IC_50_).

#### 2.8.2. In Vitro Anti-Melanogenic Properties

The cellular melanin content was evaluated using B16 4A5 melanoma cells, as detailed by Rodrigues et al. [32]. Cells were plated at 3.5 × 10^4^ into 12-well plates and allowed to adhere overnight. Then, extracts and positive control (arbutin) were applied at a concentration of 100 µg/mL (allowed cell viability higher than 80 %) for 72 h. After treatment, absorbance of the samples was measured (Biochrom EZ Read 400) and the melanin content was calculated using a standard curve of synthetic melanin (0–25 μg/mL).

#### 2.8.3. In Vitro Cytotoxicity

Cells were plated at a concentration of 1 × 10^4^ cells/well (RAW 264.7), 5 × 10^3^ cells/well (HEK 293 and HepG2), and 2 × 10^3^ cells/well (B16 4A5) in 96-well tissue plates and left to adhere overnight. Afterwards, extracts (100 µL) were applied at a concentration of 100 µg/mL for 72 h. An MTT colorimetric assay was used for cellular viability determination (Biochrom EZ Read 400), as previously described [33], and results were expressed in terms of cellular viability (%) in relation to a control containing the respective medium. 

### 2.9. Statistical Analyses

Results were expressed as mean ± standard error of the mean (SEM), and experiments were conducted at least in triplicate. Significant differences were assessed by analysis of variance (ANOVA) followed by the Tukey HSD test (*p* < 0.05). All statistical analyses were performed using the XLSTAT statistical package for Microsoft Excel (version 2013, Microsoft Corporation). The IC_50_ values were calculated by the sigmoidal fitting of data using the GraphPad Prism v. 5.0 software.

## 3. Results and Discussion

All sea lavender plants, from all the irrigation conditions, survived until the end of the experiment. However, plants irrigated with 600 mM NaCl were not able to produce flower stems and flowers [28]. Thus, data on the in vitro antioxidant activity and the metabolomic profile of these plant organs cultivated under this condition were not possible to obtain.

Extraction is the first stage for obtaining natural products from raw ingredients. Solvent extraction is the most frequently used method, but common organic solvents, such as methanol, ethanol, and dichloromethane, have many limitations regarding their ecological and toxicity roles for food, pharmaceutical, or cosmetic applications. Thus, more eco-friendly and less harmful solvents and methodologies are preferred. The use of water as an alternative extraction solvent has a significant advantage over organic solvents, such as ethanol, previously used in the extraction of natural compounds from sea lavender. Regardless of extracting different compounds than organic solvents, water extraction may generate higher yields of bioactive molecules, besides having a reduced environmental impact and no hazard, and require simple extraction equipment [34]. Moreover, ultrasound-assisted extraction (UAE) has been increasingly used for extracting bioactive compounds from natural products for food and pharmaceuticals, since it is considered a more sustainable technique with higher levels of efficiency (lower solvent, time, and energy consumption). In this context, ultrasound-assisted water extraction was used to obtain a hydrophilic extract from sea lavender biomass as a more sustainable, efficient, and safe extraction procedure. 

### 3.1. Metabolomic Profile

The metabolomic profile of the water extracts of sea lavender was established by LC-ESI-HRMS/MS, and the results are summarized in Table 1. The proposed compounds were identified based on their accurate *m*/*z* values released as deprotonated molecules [M-H]-, considering the accuracy and precision of measurement parameters such as error (ppm) and mSigma. The molecular formula was validated by extracting ionic chromatograms from the raw data, and accurate masses, isotopic patterns, and fragmentation paths were evaluated, supporting the respective chemical structures. A total of 81 compounds were tentatively identified in the water extracts of the sea lavender organs and, generally, the flowers showed higher chemical diversity, followed by leaves and peduncles (52, 47, and 28 compounds, respectively). Besides, several compounds were only identified in a specific plant organ or irrigation condition.

Twenty tree compounds, corresponding to 28% of the identified compounds, were only detected in flowers, namely, 2,3-dehydro-2-deoxy-N-acetylneuraminic acid (5), gallic acid (9), an isomer of glucosyringic acid sulphate isomer (15), aralidioside (17), glucosyl-caffeic acid sulphate (19), theasinensin B (21), prodelphinidin A2 3′-gallate (25), cis-3-hexenyl-b-primeveroside (29), licoagroside B (31), methyl licoagroside B (38), myricitin-3-O-glucoside (42), kaempferol-galloyl-hexoside (49), syringic acid (50), naringenin-7-O-glucoside (58) apigenin-7-O-glucuroide (59), prunin-6″-O-gallate (61), quercetin-3-O-acetyl-rhamnoside (65), apigenin derivative (66) 2-hydroxynaringenin (69), luteolin (72), dihydrokaempferol (74), apigenin (76), and naringenin (77). In turn, 21 compounds (26% from total compounds) were only identified in leaves, namely, a non-identified compound (1), sucrose (2), ascorbic acid sulphate (3), a salicylic acid glucosyl sulphate (8), glucosyl methyl gallate sulphate (11), glucosyl coumaric acid disulphate (14), salicylic acid glucoside (16), hydroxyferuloylglucose (36), quercetin-3-O-rutinoside (37), acetosyringone sulphate (43), two isomers of myricetin-3-O-pentoside-gallate (44, 51), myricetin-3-O-(caffeic acid-glucoside) (45), myricetin-3-O-(6-acetylgalactoside) (46), two isomers of 2′-C-methyl-myricetin-3-O-rhamnoside-gallate (47, 52), myricetin-3-O-acetyl-deoxyhexose (60), two isomers of 3′,4′,5′-trimethoxyflavanone sulphate (62, 68), medioresinol sulphate (64), and an isomer of syringaresinol sulphate (69). Five compounds (6% from total) were only present in peduncles, namely, one galloyl glucose derivative (26), di-galloyl hexose malic acid (48), tryptophan (53), myricetin-3-O-acetyl-malonyl-deoxyhexose (56), and one pinoresinol derivative (57).

**Table 1 foods-10-03104-t001:** Liquid chromatography-tandem high-resolution mass spectrometry (LC-ESI-HRMS/MS) identification of the main metabolites present in water extracts of sea lavender (*L. algarvense*) organs (flowers, peduncles and leaves) irrigated with freshwater (FWt) and two dilutions of aquaculture wastewater, corresponding to 300 and 600 mM NaCl. For distinguishing amongst very low, low, medium, and high abundance, the symbols +, ++, +++, and ++++ were used, respectively.

**Id**	**Rt** **(min)**	**Proposed Structure**	**[M-H]^−^** **[*m/z* (∆ ppm)]**	**MS/MS** **[*(m/z*) (∆ ppm) (Attribution) (%)]**	**Proposed Compound**	**Flowers**		**Peduncles**		**Leaves**		
						**FWt**	**300 mM**	**FWt**	**300 mM**	**FWt**	**300 mM**	**600 mM**
**1**	2.9	-	272.9591	158.9782	n.i.	-	-	-	-	++	+++	++++
**2**	3.1	C_12_H_22_O_11_	341.1094 (−1.3)	179.0558 (+1.8) [C_6_H_11_O_6_]^−^ (100) 119.0333 (+14.2) [C_4_H_7_O]^−^ (70)	Sucrose or isomers	-	-	-	-	++	+++	++++
**3**	3.3	C_6_H_7_O_6_SO_3_	254.9820 (−1.4)	175.0245 (+1.8) [C_6_H_7_O_6_]^−^ (100) 115.0022 (+12) [C_4_H_3_O_4_]^−^ (80)	Ascorbic acid sulphate	-	-	-	-	++++	+++	+++
**4**	3.4	C_6_H_6_O_7_	189.0032 (+4.3)	189.0034 (4.5) [C_6_H_5_O_7_]^−^ (100) 127.0046 (−7.2) [C_5_H_3_O_4_]^−^ (50)	Oxalosuccinic acid	++++	+++	++++	+++	+	++	++
**5**	3.6	C_11_H_17_NO_8_	290.0885 (−1.3)	170.0445 (8.5) [C_7_H_8_NO_4_]^−^ (10) 128.0361 (−6.4) [C_5_H_6_NO_3_]^−^ (100)	2-Deoxy-2,3-dehydro N-acetylneuraminic acid	-	++++	-	-	-	-	-
**6**	3.7	C_6_H_8_O_7_	191.0187 (+5.4)	111.0088 (−9.9) [C_5_H_3_O_4_]^−^ (100)	Citric acid	++	++	-	-	+++	+++	+++
**7**	3.8	C_13_H_10_O_8_	293.0339 (−2.2)	169.0133 (−5.6) [C_7_H_5_O_5_]^−^ (100) 137.0220 (−12.6) [C_7_H_5_O_3_]^−^ (7−0) 125.0261 (−10.2) [C_6_H_5_O_3_]^−^ (40)	Pyrogallol gallate	+	+	-	-	-	-	-
**8**	3.9	C_13_H_16_O_8_SO_3_	379.0348 (−2.1)	299.0765 (+2.4) [C_13_H_15_O_8_]^−^ (10) 241.0023 (+0.4) [C_6_H_9_O_8_S]^−^ (100)	Salicylic acid glucosyl sulphate	-	-	-	-	-	++	+
**9**	3.9	C_7_H_6_O_5_	169.0135(−5.0)	125.0241 (−6.4) [C_6_H_5_O_3_]^−^ (100)	Gallic acid	++	++	-	-	-	-	-
**10**	4.0	C_13_H_16_O_10_SO_3_	411.0235 (+0.9)	331.0671 (−5.4) [C_13_H_15_O_10_]^−^ (10) 241.0023 (+0.5) [C_6_H_9_O_8_S]^−^ (100) 169.0134 (−4.9) [C_7_H_5_O_5_]^−^ (20)	Glucogallin sulphate	++	++	++++	+++	+++	++++	++++
**11**	4.8	C_14_H_18_O_10_SO_3_	425.0398 (−0.6)	345.0828 (−0.3) [C_14_H_17_O_10_]^−^ (7) 241.0023 (+0.5) [C_6_H_9_O_8_S]^−^ (100) 183.0299(+0.1) [C_8_H_7_O_5_]^−^ (50)	Glucosyl methyl gallate sulphate	-	-	-	-	-	+	++
**12**	5.0	C_15_H_20_O_10_(SO_3_)_2_	519.01914 (−0.6)	439.0552 (−0.3) [C_15_H_19_O_13_S]^−^ (30) 241.0025 (−0.4) [C_6_H_9_O_8_S]^−^ (100)	Glucosyringic acid disulphate	++	+	-	-	++++	++++	+++
**13**	5.0	C_15_H_20_O_10_SO_3_	439.0554 (−0.4)	359.1010 (−8.1) [C_15_H_19_O_10_]^−^ (10) 241.0023 (−0.5) [C_6_H_9_O_8_S]^−^ (100) 197.0447 (−4.3) [C_9_H_9_O_5_] (10)	Glucosyringic acid sulphate	++	++	++++	++++	++++	++++	++++
**14**	5.4	C_15_H_18_O_8_(SO_3_)_2_	485.0062 (−0.6)	405.0498 (+0.2) [C_15_H_17_O_15_S]^−^ (90) 325.0928 (+0.2) [C_15_H_17_O_8_]^−^ (10)	Glucosyl coumaric acid disulphate	-	-	-	-	++	++	+++
**15**	5.4	C_15_H_20_O_10_SO_3_	439.0558 (−0.6)	359.1010 (−8.1) [C_15_H_19_O_10_]^−^ (20) 241.0026 (−0.8) [C_6_H_9_O_8_S]^−^ (100)	Glucosyringic acid sulphate isomer	+++	+++	-	-	-	-	-
**16**	5.8	C_13_H_16_O_16_SO_3_	379.0348 (−2.1)	299.0768 (+1.4) [C_13_H_15_O_8_]^−^ (15) 241.0023 (−0.5) [C_6_H_9_O_8_S]^−^ (100)	Salicylic acid glucoside	-	-	-	-	+	-	+
**17**	5.9	C_18_H_24_O_13_	447.1144 (−1.0)	429.1041 (−0.6) [C_18_H_21_O_12_]^−^ (60) 339.0722 (−0.1) [C_15_H_15_O_9_]^−^ (30) 301.0568 (−1.1) [C_12_H_13_O_9_]^−^ (20)	Aralidioside	++	++	-	-	-	-	-
**18**	5.9	C_15_H_18_O_8_SO_3_	405.0496 (+0.1)	325.0926 (+0.8) [C_15_H_17_O_8_]^−^ (10) 241.0025 (−0.4) [C_6_H_9_O_8_S]^−^ (100) 163.0396 (+2.2) [C_9_H_7_O_3_]^−^ (10)	Glucosyl coumaric acid sulphate	++	+++	-	-	+++	++++	++++
**19**	6.2	C_15_H_18_O_9_SO_3_	421.0451 (−1.1)	341.0660 (−1.3) [C_15_H_17_O_9_]^−^ (7) 241.0018 (+0.5) [C_6_H_9_O_8_S]^−^ (100) 179.0344 (+3.4) [C_9_H_7_O_4_]^−^ (40)	Glucosyl-caffeic acid sulphate	+	+	-	-	-	-	-
**20**	6.4	C_20_H_20_O_14_	483.0779 (+0.3)	313.0671 (−0.1) [C_13_H_15_O_10_]^−^ (20) 313.0564 (+0.3) [C_13_H_13_O_9_]^−^ (60) 271.0461 (−0.5) [C_11_H_11_O_8_]^−^ (100)169.0129 (+8.1) [C_7_H_5_O_5_]^−^ (40)	Digalloyl glucose	+	++	++	++	-	-	-
**21**	6.5	C_37_H_30_O_18_	761.1356 (0.4)	609.1249 (+0.1) [C_30_H_25_O_14_]^−^ (20) 423.0721 (+0.0) [C_22_H_15_O_9_]^−^ (100) 305.0667 (+0.0) [C_15_H_13_O_7_]^−^ (70)169.0135 (−3.5) [C_7_H_5_O_5_]^−^ (7)	Theasinensin B	-	+++	-	-	-	-	-
**22**	6.5	C_15_H_18_O_8_SO_3_	405.0496 (−1.6)	241.0025 (−0.4) [C_6_H_9_O_8_S]^−^ (100) 163.0395 (+3.6) [C_9_H_7_O_3_]^−^ (30)	Glucosyl coumaric acid sulphate isomer	++	-	++	+++	-	-	-
**23**	6.8	C_15_H_18_O_8_(SO_3_)_2_	485.0061 (+0.7)	405.0497 (+0.1) [C_15_H_17_O_11_S]^−^ (40) 325.0928 (+0.2) [C_15_H_17_O_8_]^−^ (5) 241.0024 (−0.1) [C_6_H_9_O_8_S]^−^ (100)	Glucosyl coumaric acid disulphate isomer	+++	++++	-	-	+++	+++	++++
**24**	6.8	C_15_H_18_O_8_SO_3_	405.0498 (−0.3)	325.0930 (−0.5) (C_15_H_17_O_8_)^−^ (10) 241.0023 (−1.8) [C_6_H_9_O_8_S]^−^ (100) 163.0397 (2.5) [C_9_H_7_O_3_]^−^ (10)	Glucosyl coumaric acid sulphate isomer	+++	++	+++	+++	-	++++	++++
**25**	7.0	C_37_H_28_O_18_	759.1207 (−0.6)	423.0724 (−0.6) [C_22_H_15_O_9_]^−^ (100) 301.0354 (−0.2) [C_15_H_9_O_7_]^−^ (80)	ProdelphinidinA2 3′-gallate	++++	+++	-	-	-	-	-
**26**	7.0	C_20_H_22_O_12_	453.1040 (−0.3)	313.0567 (+0.7) [C_13_H_13_O_9_]^−^ (100) 169.0131 (+6.5) [C_7_H_5_O_5_]^−^ (40)	Galloyl glucose derivative	-	-	+++	+++	-	-	-
**27**	7.4	C_11_H_14_O_4_SO_3_	289.0393 (−1.8)	209.0826 (−3.1) [C_11_H_13_O_4_]^−^ (60) 149.0599 (5.9) [C_9_H_9_O_2_]^−^ (100)	Sinapyl alcohol sulphate	++++	++++	++++	++++	++++	++++	++++
**28**	7.5	C_15_H_16_O_10_SO_3_	435.0240 (−0.2)	355.0671 (−0.1) [C_15_H_15_O_10_]^−^ (20) 197.0444 (+6.0) [C_9_H_9_O_5_]^−^ (100)	Caffeic acid-3-glucuronide sulphate	++	++	-	-	++++	+++	++++
**29**	7.7	C_17_H_30_O_10_	393.1770 (−0.9)	271.0610 (+0.8) [C_15_H_11_O_5_]^−^ (20) 205.0709 (+4.3) [C_8_H_13_O_6_]^−^ (100) 119.0337 (+7.8) [C_4_H_7_O_4_]^−^ (80)	Cis-3-hexenyl-b-primeveroside	++++	++++	-	-	-	-	-
**30**	7.8	C_28_H_24_O_17_	631.0944 (−0.6)	479.0830 (+0.2) [C_21_H_19_O_13_]^−^ (90) 316.019 (−1.8) [Y_0_−H]^−^^∙^ [C_15_H_8_O_8_]^−^^∙^ (100)	Myricetin-3-O-galloyl-hexoside	+++	+++	+++	+++	++	+++	++++
**31**	8.0	C_18_H_24_O_12_	431.1192 (−0.2)	285.0625 (−3.1) [C_12_H_13_O_8_]^−^ (10) 225.0409 (−1.8) [C_10_H_9_O_8_]^−^ (100)	Licoagroside B	+	+++	-	-	-	-	-
**32**	8.0	C_27_H_30_O_17_	625.1418 (−1.2)	316.0233 (−2.8) [Y_0_−H]^−^^∙^ [C_15_H_8_O_8_]^−^^∙^ (100)	Myricetin-3-O-rutinoside	-	-	++	++	++	+++	+++
**33**	8.2	C_21_H_20_O_13_	479.0835 (+0.1)	316.023 (−0.9) [Y_0_−H]^−^^∙^ [C_15_H_8_O_8_]^−^^∙^ (100) 271.0253 (−1.9) (C_14_H_7_O_6_)^−^ (20)	Myricetin-3-O-glucoside	+++	+++	-	-	++	+++	++++
**34**	8.5	C_22_H_22_O_12_	477.1036 (+0.4)	433.1145 (−1.1) (C_21_H_21_O_10_)^−^ (20) 313.0567 (−0.7) (C_13_H_13_O_9_)^−^ (100) 169.0141 (+1.0) [C_7_H_5_O_5_]^−^ (40)	Galloylhexoside derivative	++	++	++	+++	-	-	-
**35**	8.6	C_28_H_24_O_16_	615.0985 (+0.1)	463.0882 (+0.1) [C_21_H_19_O_12_]^−^ (100) 301.0345 (+3.0) [Y_0_]^−^ [C_15_H_9_O_7_]^−^ (80) 300.0279 (−1.0) [Y_0_−H]^−^^∙^ [C_15_H_8_O_7_]^−^^∙^ (90)	Quercetin-3-O-galloyl-hexoside	-	-	-	+	+	+	+
**36**	8.7	C_18_H_20_O_10_	371.0981 (+0.6)	249.0615 (+0.2) [C_9_H_13_O_8_]^−^ (100)	Hydroxyferuloylglucose	-	-	-	-	-	+++	+++
**37**	8.7	C_27_H_30_O_16_	609.1456 (+0.8)	463.0882 (−0.1) [C_22_H_19_O_12_]^−^ (400) 300.0275 (−0.8) [Y_0_−H]^−^^∙^ [C_15_H_8_O_7_]^−^^∙^ (100)	Quercetin-3-O-rutinoside	-	-	-	-	++	++	+++
**38**	8.8	C_19_H_26_O_12_	445.1350 (+0.2)	285.0623 (−2.6) [C_12_H_13_O_8_]^−^ (7) 225.0429 (−1.9) [C_10_H_9_O_8_]^−^ (100)	Methyl licoagroside B	+++	+++	-	-	-	-	-
**39**	8.8	C_14_H_14_O_9_ SO_3_	405.0145 (−1.2)	325.0564 (0.3) [C_14_H_13_O_9_]^−^ (20) 209.0454 (+0.5) [C_10_H_9_O_5_]^−^ (50) 167.0344 (+3.2) [C_8_H_7_O_4_]^−^ (100)	Galloylshikimic acid sulphate	-	++++	++++	++++	++++	++++	++++
**40**	9.0	C_21_H_20_O_12_	463.0872 (+2.2)	316.020 (−1.8) [Y_0_−H]^−^^∙^ [C_15_H_8_O_8_]^−^^∙^ (100) 217.0250 (−0.8) [C_14_H_7_O_6_]^−^ (20)	Myricetin-3-O-rhamnoside	-	++++	++	++	++	+++	++++
**41**	9.0	C_15_H_16_O_10_SO_3_	435.0251 (−2.9)	355.0677 (−1.7) [C_15_H_15_O_10_]^−^ (10) 197.0455 (+5.5) [C_9_H_9_O_5_]^−^ (100)	Caffeic acid glucuronic sulphate	++++	++++	++++	++++	++++	++++	++++
**42**	9.3	C_21_H_20_O_13_	479.0835 (+0.1)	316.023 (−0.9) [Y_0_−H]^−^^∙^ [C_15_H_8_O_8_]^−^^∙^ (100)	Myricetin-3-O-glucoside	+	+	-	-	-	-	-
**43**	9.3	C_20_H_24_O_8_SO_3_	471.0967 (−0.5)	275.0228 +(0.9) [C_10_H_11_O_4_SO_3_]^−^ (100) 195.0651 (+6.1) [C_10_H_11_O_4_]^−^ (50)	Acetosyringone sulphate	-	-	-	-	++	++	++
**44**	9.3	C_28_H_24_O_15_	599.1036 (+0.7)	316.022 (−0.8) [Y_0_−H]^−^^∙^ [C_15_H_8_O_8_]^−^^∙^ (100)	Myricetin-3-O-pentoside-gallate	-	-	-	-	-	-	+
**45**	9.3	C_30_H_28_O_17_	659.1244 (+1.5)	316.019 (−1.7) [Y_0_−H]^−^^∙^ [C_15_H_8_O_8_]^−^^∙^ (100) 479.0811 (+4.2) [C_21_H_19_O_13_]^−^ (20)	Myricetin-3-O-(3-caffeic acid-glucoside)	-	-	-	-	+	++	+++
**46**	9.4	C_23_H_22_O_14_	521.0931 (−1.1)	316.023 (−0.9) [Y_0_−H]^−^^∙^ [C_15_H_8_O_8_]^−^^∙^ (100)	Myricetin 3-O-(6-acetylgalactoside)	-	-			+	++	+++
**47**	9.4	C_29_H_26_O_16_	629.1143 (+0.9)	316.023 (−0.9) [Y_0_−H]^−^^∙^ [C_15_H_8_O_8_]^−^^∙^ (100)	2’-C-methyl-myricetin-3-O-rhamnoside-gallate	-	-	-	-	-	+	++
**48**	9.4	C_22_H_26_O_8_SO_3_	599.1042 (+0.8)	447.0932 (+0.2) [C_21_H_19_O_11_]^−^ (30) 313.0563 (+0.5) [C_13_H_13_O_9_]^−^ (100) 169.0141 (+1.0) [C_7_H_5_O_5_]^−^ (40)	Di-galloyl-hexose malic acid	-	-	+	+	-	-	-
**49**	9.4	C_28_H_24_O_15_	599.1045 (−0.4)	447.0943 (−0.8) [C_21_H_19_O_11_]^−^ (50) 285.0404 (0.1) [Y_0_−H]^−^^∙^ [C_15_H_9_O_6_]^−^ (100)	Kaempferol-galloyl-hexoside	-	++	-	-	-	-	-
**50**	9.5	C_9_H_10_O_5_	197.0445 (+5.5)	124.0166 (−0.4) [C_6_H_4_O_3_]^−^ (100)	Syringic acid	+++	-	-	-	-	-	-
**51**	9.5	C_28_H_24_O_15_	599.1037 (+0.9)	447.0931 (+0.3) [C_21_H_19_O_11_]^−^ (30) 316.022 (−0.8) [Y_0_−H]^−^^∙^ [C_15_H_8_O_8_]^−^^∙^ (100)	Myricetin-3-O-pentoside-gallate isomer	-	-	-	-	++	++	++
**52**	9.6	C_29_H_26_O_16_	629.1143 (+0.9)	316.023 (−0.9) [Y_0_−H]^−^^∙^ [C_15_H_8_O_8_]^−^^∙^ (100)	2′-C-methyl-myricetin-3-O-rhamnoside-gallate isomer	-	-	-	-	+	++	-
**53**	9.8	C_13_H_14_N_2_O_3_	203.0818 (+4.0)	142.0648 (+9.2) [C_10_H_8_N]^−^ (40)	Tryptophan	-	-	++++	++	-	-	-
**54**	9.8	C_13_H_14_N_2_O_3_	245.0932 (+0.8)	203.0818 (+4.0) [C_11_H_11_N_2_O_2_]^−^ (100) 142.0648 (+9.2) [C_10_H_8_N]^−^ (40)	N-acetyl-tryptophan	++++	+	-	-	++++	+++	+++
**55**	9.8	C_14_H_14_N_2_O_5_	289.0830 (−0.1)	203.0818 (+4.0) [C_11_H_11_N_2_O_2_]^−^ (100)	N-manoyl-tryptophan	+++	+++	+++	-	+++	++	++
**56**	9.8	C_24_H_22_O_5_	549.0891 (−1.0)	316.025 (−2.2) [Y_0_−H]^−^^∙^ [C_15_H_8_O_8_]^−^^∙^ (100) 217.0250 (−0.8) [C_14_H_7_O_6_]^−^ (20)	Myricetin-3-O-acetyl--malonyl-deoxyhexose	-	-	-	+	-	-	-
**57**	10.0	C_27_H_36_O_12_	551.2134 (+0.0)	357.1346 (−0.7) [C_20_H_12_O_6_]^−^ (100)	Pinoresinol derivative	-	-	+	-	-	-	-
**58**	10.1	C_21_H_22_O_10_	433.1136 (+1.0)	271.0610 (−0.1) [Y_0_]^−^ [C_15_H_11_O_5_]^−^ (100)	Naringenin-7-O-glucoside	++	++	-	-	-	-	-
**59**	10.1	C_21_H_18_O_11_	445.0775 (+0.3)	269.0456 (−0.1) [Y_0_]^−^ [C_15_H_9_O_5_]^−^ (100)	Apigenin-7-O-glucuronide	++++	+++	-	-	-	-	-
**60**	10.2	C_23_H_22_O_13_	505.0995 (−1.2)	316.0230 (−0.9)[Y_0_−H]^−^^∙^ [C_15_H_8_O_8_]^−^, (100) 217.0253 (−1.9) [C_14_H_7_O_6_]^−^ (15)	Myricetin-3-O-acetyl-deoxyhexose	-	-	-	-	+++	+++	++++
**61**	10.5	C_28_H_26_O_14_	585.1247 (+0.4)	439.0880 (+0.5) [C_19_H_19_O_13_]^−^ (100) 271.0608 (+1.2) [Y_0_]^−^ [C_15_H_11_O_5_]^−^ (10)	Prunin-6′’-O-gallate	+++	+++	-	-			
**62**	10.5	C_18_H_18_O_7_SO_3_	425.0545 (+0.3)	345.0980 (+0.0) [C_18_H_17_O_7_]^−^ (40) 315.0872 (+0.5) [C_17_H_15_O_6_]^−^ (60) 300.0638 (+0.6) [C_16_H_12_O_6_]^−^^∙^ (100)	3′,4′,5′-Trimethoxyflavanone sulphate		-	-	-	+++	+++	+++
**63**	10.6	C_22_H_26_O_8_SO_3_	497.1125 (−0.3)	417.1556 (−0.3) [C_22_H_25_O_8_]^−^ (100) 181.0493 (+7.4) [C_9_H_9_O_4_]^−^ (90)	Syringaresinol sulphate	++	+++	++	++	+++	+++	++++
**64**	10.7	C_17_H_24_O_7_SO_3_	467.1019 (+1.1)	387.1448 (+0.4) [C_17_H_23_O_7_]^−^ (100) 372.1212 (+0.5) [C_16_H_20_O_7_]^−^^∙^ (90) 357.0978 (+0.4) [C_19_H_17_O_7_]^−^ (40) 181.0494 (+7.0) [C_9_H_9_O_4_]^−^ (50) 151.0309 (+6.8) [C_8_H_7_O_3_]^−^ (40)	Medioresinol sulphate	-	-	-	-	++	+++	+++
**65**	10.7	C_23_H_22_O_12_	489.1042 (−0.8)	300.0280 (−1.6) [Y_0_−H]^−^^∙^ [C_15_H_8_O_7_]^−^^∙^ (100) 271.0153 (−1.9) [C_14_H_7_O_6_]^−^ (30)	Quercetin-3-O-acetyl-rhamnoside	+	++	-	-	-	-	-
**66**	11.0	C_29_H_26_O_14_	597.1245 (−0.5)	413.0887 (−2.2) [C_21_H_17_O_9_]^−^ (10) 301.03607 (−1.9) [C_15_H_9_O_7_]^−^ (20) 269.0456 (−0.3) [Y_0_]^−^ [C_15_H_9_O_5_]^−^^∙^ (100)	Apigenin derivative	+	++	-	-	-	-	-
**67**	11.2	C_15_H_12_O_6_	287.0563 (+0.6)	151.0031 (+3.9) [^1,3^A^−^] [C_7_H_3_O_4_]^−^ (20) 135.0435 (−0.8) [^1,3^B^−^] [C_8_H_7_O_2_]^−^ (100)	2-Hydroxynaringenin	++	++	-	-	-	-	-
**68**	11.3	C_18_H_18_O_7_SO_3_	425.0553 (+0.9)	-	3′,4′,5′-Trimethoxyflavanone sulphate isomer	-	-	-	-	++	++	++
**69**	11.5	C_22_H_26_O_8_SO_3_	497.1135 (−2.3)	417.1164 (−2.1) [C_22_H_25_O_8_]^−^ (50) 402.1326 (−1.5) [C_21_H_22_O_8_]^−^ (90) 387.1093 (−2.0) [C_20_H_19_O_8_]^−^ (60) 181.0500 (+3.4) [C_9_H_9_O_4_]^−^ (100)	Syringaresinol sulphate isomer	-	-	-	-	+	++	++
**70**	11.5	C_20_H_22_O_6_SO_3_	437.0904 (+1.0)	357.1339 (+1.4) [C_20_H_12_O_6_]^−^ (100) 342.1104 (+1.3) [C_19_H_18_O_6_]^−^ (85) 151.0391 (+6.3) [C_8_H_7_O_3_] (70)	Pinoresinol sulphate	++++	+++	+++	+++	++++	++++	++++
**71**	12.1	C_20_H_22_O_6_SO_3_	437.0916 (+1.8)	357.1339 (+1.4) [C_20_H_12_O_6_]^−^ (100)	Pinoresinol sulphate isomer	++	+	-	-	+++	+++	+++
**72**	12.2	C_15_H_20_O_6_	285.0403 (+0.6)	199.0392 (+4.2) [C_12_H_7_O_3_]^−^ (50) 175.0390 (+6.1) [C_10_H_7_O_3_]^−^ (60) 151.0029 (+4.9) [^1,3^A^−^] [C_7_H_3_O_4_]^−^ (40) 133.0279 (−8.3) [^1,3^B^−^] [C_8_H_5_O_2_]^−^ (100)	Luteolin	++	++	-	-	-	-	-
**73**	12.2	C_25_H_24_O_14_	547.1102 (−1.6)	316.023 (−0.9) [Y_0_−H]^−^^∙^ [C_15_H_8_O_8_]^−^^∙^ (100)	Myricetin-3-O-diacetylrhamnoside	+	+	-	+	+++	++	++++
**74**	12.3	C_15_H_12_O_6_	287.0565 (−1.2)	269.0458 (−1.0) [C_15_H_9_O_5_]^−^ (30) 259.0614 (+0.6) [C_14_H_11_O_5_]^−^ (30) 177.0546 (+6.6) [C_10_H_9_O_3_]^−^ (100) 151.0031(+3.8) [^1,3^A^−^] [C_7_H_3_O_4_]^−^ (40)	Dihydrokaempferol	++	+++	-	-	-	-	-
**75**	13.3	C_18_H_22_O_5_	327.2175 (−0.6)	-	Trihydroxy-10,15-octadecadienoic acid	+++	+++	+++	+++	++	++	++
**76**	13.5	C_15_H_10_O_5_	269.0457 (−0.6)	227.0347 (−1.1) [C_13_H_7_O_4_]^−^ (60) 151.0030 (+2.5) [^1,3^A^−^] [C_7_H_3_O_4_]^−^ (70) 117.0324 (+9.1) [^1,3^B^−^] [C_8_H_7_O]^−^ (100)	Apigenin	+++	+	-	-	-	-	-
**77**	13.7	C_15_H_12_O_5_	271.0615 (−1.3)	187.0393 (+4.3) [C_11_H_7_O_3_]^−^ (40) 151.0030 (+3.4) [^1,3^A^−^] [C_7_H_3_O_4_]^−^ (50) 119.0490 (+9.9) [^1,3^B^−^] [C_8_H_7_O]^−^ (100)	Naringenin	+++	++++	-	-	-	-	-
**78**	14.1	C_18_H_34_O_5_	329.2332 (−0.5)	-	Trihydroxy-10-octadecenoic acid	+++	++++	++	++	++	++	++
**79**	14.3	C_13_H_24_O_3_SO_3_	307.1220(−0.2)	-	Oxo-tridecanoic acid sulphate	+	-	++++	++	++	+++	+++
**80**	14.9	C_18_H_12_O_4_	287.2227 (+0.4)	-	10, 16-Dihydroxyhexadecanoic acid	+	-	-	-	+++	++	++

-: not detected.

Additionally, the irrigation salinity influenced the relative abundance of some compounds. For instance, increasing salinity (including all tested salinities) led to a general increase in relative abundance of compounds 1, 2, 18, 20, 22, and 23. Compounds 10, glucosyl coumaryl acid sulphates (14 and 24), hydroxyferuloylglucose (36), and several myricetin glycoside derivatives (32, 40, 45, 46, 47, 60, and 73), two syringaresinol sulphates (63 and 69), and one medioresinol sulphate (64) only decreased in the leaves’ extracts. However, digalloyl glucose (20) and one galloylhexoside derivative (34) only decreased in flowers and peduncles, respectively. For some other compounds, as the irrigation salinity increased (among all tested salinities), their relative abundance decreased, namely compounds 3, 8, 12, 22, 24, syringic acid (50), tryptophan (53), and one pinoresinol sulphate isomer (71). Oxalosuccinic acid (4) decreased in both flowers and peduncles, apigenin (76) declined only in flowers, whereas for the pinoresinol derivative (57) only in peduncles.

Plants contain a wide variety of secondary metabolites, namely polyphenolic compounds, as, for example, flavonoids, tannins, and phenolic acids that are the most common plant-derived natural products, and that are widely present in the sea lavender aqueous extracts [35]. Moreover, enzymatic modifications of these known molecules result in the generation of many types of derivatives, such as prenylated, acetylated, methylated, sulphated, glucuronated, and glycosylated compounds [36]. Sulphated phenolics were usually found in plants from saline areas, which indicates a strong correlation between environments rich in salts and the biosynthesis of sulphated compounds. Despite their functional role in plants is still not being evident, these structural modifications can be considered ecological adaptations with important functions in co-pigmentation, plant growth regulation, molecular recognition, detoxification, and signalling pathways [36,37]. Moreover, sulphation, methylation, and glycosylation contribute to the improvement of the solubility, stability, and biological activities of these molecules, as, for example, negatively charged sulphated derivatives that have higher water solubility, important for interactions with biological targets [36,38,39]. Galloylation of phenolic compounds also affects their properties, which is important for their protective antioxidant mechanisms; for example, galloyl groups may increase the capacity to donate electrons, chelate iron, regenerate tocopherol, and for lipophilicity [40]. Thus, galloylation of polyphenols changes their biological properties, with more galloyl moieties in the structure resulting in increased biological activity when compared to the parent molecules [41]. Therefore, the high representativity of glycosylated, sulphated, and galloylated phenolic compounds in extracts of produced sea lavender plants may be related to stress resistance mechanisms, namely, to high UV radiation and temperature which they are subjected to in the greenhouse, as well as irrigation salinity. For example, salt stress promotes the accumulation of glucose derivatives for mitigating stress conditions, including osmoprotection, carbon storage, and scavenging free radicals, which may ultimately also affect the biological properties of the produced plants [42].

Twenty-eight of those identified have already been reported in infusions made from flowers of this species, including gallic (9) and syringic (46) acids, and apigenin (72). Furthermore, these and 24 more compounds (2, 26, 27, 30–34, 40, 46–47, 51, 54, 60, 66–67, 70, 72, 74–75, and 78–79) have also been described in ethanol extracts of sea lavender grown under saline cultivation [25,26,27,28]. Several other compounds were detected in *L. algarvense* for the first time; however, they were previously described in other *Limonium* species, as, for example, citric acid (6) detected in methanol leaf extract from *L. globuliferum* and *L. quesadense* [43,44] and quercetin-galloyl-hexoside (35) identified in *L. delicatulum* and *L. quesadense* [44]. Myricetin-3-O-glucoside (33) was previously detected in *L. caspium* and *L. aureum* [45,46], while tryptophan (53) was reported in *L. doufourii* and *L. albuferae* aerial parts grown under salt stress [47].

### 3.2. In Vitro Antioxidant Activity

In a preliminary approach to appraise the antioxidant properties of ultrasound-assisted water extracts, all samples were subjected to a preliminary evaluation by five complementary in vitro methods, including radical and redox metal assays, and the results are summarized in Table 2. Flowers’ extracts from freshwater-irrigated sea lavenders had the highest capacity to scavenge the ABTS radical, to chelate copper, and to reduce iron (IC_50_ values of 397, 642, and 129 μg/mL, respectively). However, when plants were irrigated with saline aquaculture wastewaters, the peduncles from plants watered with 300 mM NaCl had the best capacity to scavenge DPPH (IC_50_ = 383 μg/mL), whereas flowers exhibited the best activity in the ABTS, CCA, and FRAP assays (IC_50_ = 617, 720, and 191 μg/mL). None of the samples had significant ICA at the maximum concentration tested (1 mg/mL). 

As discussed in Section 3.1, the sea lavender water extracts are rich in phenolic compounds, mainly flavonoids, tannins, and phenolic acids, as well as their sulphated, glycosylated, and galloylated derivatives (Table 1). It is well-documented that phenols are great antioxidants that may act in several ways: (1) hydrogen-donating antioxidants react with reactive oxygen and nitrogen species, stopping the production of radicals by generating a molecule that is more chemically stable than the initial radical; (2) related to their metal chelating properties that participate in free radicals generation; or (3) interaction with proteins, due to their ability to inhibit enzymes implicated in free radicals formation, such as cytochrome P450, lipoxygenases, cyclooxygenase, and xanthine oxidase [35,48]. Therefore, natural products rich in these compounds, such as the water extract studied in this work, have the potential to be used as antioxidant supplements, to inactivate free radicals, and decrease the potential occurrence of cellular damage that leads to disease development [49], or as antioxidant food additives, to slow or stop the breakdown of fats and oils, thus contributing to food preservation [50].

The greater antioxidant capacity found in the flowers’ extracts may be related to the higher diversity of flavonoids and its derivatives detected in this plant organ (Table 1), since this group of molecules is well-known for this capacity; however, studies that support the activity of the most identified compounds alone are limited. Furthermore, the antioxidant activity of sea lavender water extracts generally decreased with increasing irrigation salinity. The presence of salt ions might affect plant growth and secondary metabolism, which influences the quantitative and qualitative variation of antioxidant molecules and its biological properties [20]; therefore, the decrease in the relative abundance of some of the detected compounds (stated in Section 3.1) with the increase in irrigation salinity is likely to be reflected in the reduced antioxidant activity observed. Still, these molecules are essential to reduce the levels of oxidative radicals induced by harsh environmental conditions [20]. Additionally, while the synthesis of some compounds (e.g., phenolics) can be impaired, other biosynthetic pathways can be activated by higher UV-radiation or salinity, such as pigments (e.g., chlorophylls, carotenoids) or proline [51].

Despite the few studies reporting on the impact of salinity on halophytes’ biological properties, some species display the same pattern as sea lavender. For example, acetone extracts from stems and leaves of *Polygonum maritimum* L. (sea knotgrass) showed reduced antioxidant capacity (in vitro radical-scavenging activity and copper chelation) with increasing irrigation salinity (from approx. 0 to 600 mM NaCl) when grown under greenhouse conditions [27]. The DPPH radical-scavenging activity of methanol extracts from *Sesuvium portulacastrum* (L.) (shoreline purslane or sea purslane) grown in outdoor containers was also influenced by salinity, while stems had reduced antioxidant capacity, and the leaves and roots displayed increased activity with augmented salt concentration (0 to 200 mM NaCl) [51]. Likewise, methanol extracts from greenhouse-cultivated *Cakile maritima* L. (sea rocket), obtained from seeds collected in Tabarka (Tunisia), exhibited reduced antioxidant activity when plants were irrigated with water containing higher NaCl concentrations (0–400 mM), but the same species obtained from seeds collected in Jerba (Tunisia) presented an opposite trend [52]. These data indicate that the antioxidant capacity of cultivated plants may be triggered by diverse factors, such as the local collection, the plant’s developmental stage, and plant organ, even for the same species, suggesting that many parameters may be involved and that adaptations may not be species-specific. 

As the flowers’ extracts had the highest antioxidant capacity overall, they were therefore selected from fresh- and saltwater irrigated (300 mM NaCl) plants to be further appraised for their potential use as an ingredient source for herbal supplements and food additives. For that, extracts were analyzed for their ex vivo antioxidant, anti-melanogenic, and anti-inflammatory properties and toxicity, and for in vitro tyrosinase inhibition.

### 3.3. Ex Vivo Antioxidant Activity

Most methods for evaluating antioxidant activity are performed in vitro and based on radicals with reduced or nil biological relevance. Such in vitro methods are useful for screening purposes, but the obtained results should be confirmed in assays with biological targets that resemble those found in vivo, such as ex vivo methods, where cells are taken from an in vivo model and are used in vitro tests by using radicals and substrate targets with higher biological relevance compared to conventional assays. 

In this work, the selected extracts were tested by two ex vivo antioxidant assays, which allowed to evaluate their ability to inhibit lipid peroxidation (by the TBARS formation) and oxidative haemolysis (OxHLIA), as shown in Table 3. The flower extract from plants irrigated with the aquaculture wastewater containing 300 mM NaCl was the most active (IC_50_ = 81 µg/mL) in TBARS compared to that from freshwater-irrigated plants (IC_50_ = 127 µg/mL). In the OxHLIA assay, no significant difference was found amongst the two irrigation systems (freshwater: IC_50_ = 136; 300 mM NaCl: IC_50_ = 140 µg/mL). To our best knowledge, this is the first report on the use of ex vivo assays to determine the antioxidant activity of sea lavender. 

Lipid peroxidation is associated with several degenerative disorders due to the high susceptibility of lipids to oxidation. Moreover, the integrity of cell membranes is kept by lipids; thus, extensive lipid peroxidation alters their composition, structure, and function, promoting further damage of DNA and proteins that ultimately causes cellular death. Reducing cellular lipid peroxidation may be crucial to prevent the occurrence of degenerative and chronic disorders linked to oxidative stress. Furthermore, lipid peroxidation is also a major cause of deterioration in foods rich in fat, especially those containing polyunsaturated fats (PUFAs). BHT, BHA, and α-tocopherol are common lipid-soluble antioxidants used by food industries to prevent oxidation. However, they are volatile and decompose at high temperatures, and are also associated with several health concerns, such as skin allergic reactions, carcinogenic effects, and hormonal dysregulation [53,54,55]. In addition, they are toxic to aquatic organisms and have the potential to bioaccumulate [56]. Thus, the use of halophytes as antioxidants in foods is a promising alternative to the use of those of synthetic origin, mainly due to the increasing request of consumers for natural food additives [57,58]. Thus, TBARS and OxHLIA assays are good ex vivo models for evaluating inhibition of lipid peroxidation by the presence of antioxidants [59,60]. As discussed in former sections, sea lavender flower extracts’ richness in flavonoids and its derivatives may be related to the high antioxidant activity of these samples. This makes these extracts great candidates to be used as ingredients in herbal supplementation products with the aim of improving general health and well-being, and/or as food additives for preventing lipid oxidation of lipid-rich foods.

### 3.4. Tyrosinase Inhibition

Tyrosinase is a multi-copper enzyme with a key role in melanin biosynthesis and enzymatic browning. Excessive melanin production and accumulation occur in several types of skin diseases, such as melasma, periorbital hyperpigmentation, and lentigines, and is also linked with an increased risk of skin cancer and with neurodegenerative ailments, including Parkinson’s disease [61,62,63,64]. The demand for new tyrosinase inhibitors from natural sources is on the rise due to the problems presented by some tyrosinase inhibitors currently in use, such as hydroquinone, which is potentially mutagenic to mammalian cells and linked to several adverse reactions (e.g., contact dermatitis, transient erythema) and arbutin, which is chemically unstable [65,66]. Enzymatic browning is a major problem of fresh-cut fruits, which results from oxidation reactions with several enzymes and leads to modifications in the appearance of the nutritional value of foodstuffs [5,6,7]. Sulfiting agents are the most frequently used anti-browning products, but have adverse health effects [8,9]. Thus, safer anti-browning additives are much needed, and several natural products were already identified, including polyphenol-rich extracts [10,67]. In this work, selected extracts were tested for tyrosinase inhibition, and results are depicted on Table 3. The flower extracts from plants irrigated with 300 mM NaCl allowed for the lowest IC_50_ value (873 µg/mL), while those from freshwater-irrigated plants were not active up to the concentration of 1000 µg/mL. However, other *Limonium* species have showed lower IC_50_ values, namely, *L. delicatulum* methanol extracts from roots (9.87 µg/mL) and leaves (24.77 µg/mL) [68], as well as hexane and ethyl acetate fractions from *L. effusum* methanol extracts (148–295 µg/mL) [69].

A high number of natural tyrosinase inhibitors are phenolics, including flavonoids and their derivatives. Some synthetic inhibitors are based on natural flavonoid skeletons providing an effective scaffold for the development of novel tyrosinase inhibitors. For example, flavonoids containing a keto group (e.g., kaempferol and quercetin) are described to have potent tyrosinase inhibition due to their capacity to chelate cooper in the enzyme active site [70]. In this sense, the higher tyrosinase inhibition found in flower extracts irrigated with 300 mM NaCl may be related with the high abundance of flavonoid-containing keto groups in this extract, such as dihydrokaempferol, naringenin, apigenin, and some of their derivatives. Additionally, other studies indicate that the number and location of the hydroxyl group on the flavonoids affect their inhibitory capacity towards tyrosinase. For example, the number of hydroxyl groups on the B ring of flavonoids or catechins improves their tyrosinase inhibition, which may also be correlated with their enhanced antioxidant activity [71]. Thus, the occurrence of phenolics with multiple hydroxyl groups, such as theasinensin B, licoagroside B, myricetin-3-O-rhamnoside, galloylshikimic acid sulphate, kaempferol-galloyl-hexoside, quercetin-3-O-acetyl-rhamnoside, and dihydrokaempferol, found in higher abundance in the flower extract from plants irrigated with 300 mM NaCl, can be related with the higher tyrosinase inhibition observed. Furthermore, since copper is a cofactor of the tyrosinase enzyme, the presence of copper-chelating compounds can lead to enzyme inactivation. Thus, the inhibitory activity of this extract may also be associated with its high antioxidant and copper chelating capacity. Overall, our results suggest that the flower extracts from sea lavender irrigated with saline aquaculture wastewaters contain molecules with tyrosinase inhibitory properties, most probably phenolic acids and flavonoids, and therefore with interest in the cosmetic, pharmaceutical and food industries.

### 3.5. In Vitro Anti-Inflammatory Properties

A wide range of mental and physical health disorders involve inflammation, such as chronic inflammatory states (e.g., ischemic heart disease, stroke, cancer, diabetes mellitus, chronic kidney disease, non-alcoholic fatty liver disease, autoimmune and neurodegenerative conditions) and have been recognized to comprise 50% of all deaths worldwide [72]. In this work, RAW 264.7 macrophages were stimulated with LPS to produce nitric oxide (NO) to simulate chronic inflammation [73], a common method for assessing the anti-inflammatory potential of botanicals [33]. However, none of the tested samples showed the ability to reduce the LPS-induced NO production when tested up to 400 µg/mL (data not shown). However, previous studies have reported the capacity of aqueous extracts of *L. algarvense* flowers collected from the wild to reduce the NO production in RAW 264.7 cells (IC_50_ = 46–48 µg/mL) [26]. The probable reason for this discrepancy could be the biomass origin and extraction methodology, since the reported activity was detected in plants collected from the wild extracted by infusion (100 °C for 5 min) instead of plants produced in a greenhouse extracted by ultrasounds for 30 min. This variation of chemical and biological properties is often noticed by other authors, and usually attributed to different samples’ origin, divergent extraction methodologies, and/or interspecific variability [74].

### 3.6. Toxicological Evaluation

Although natural products are generally considered safer that its synthetic counterparts, it is well-known that “natural” does not necessary mean “safe” [75]. Thus, ensuring the toxicological safety of herbal ingredients is of major concern. Therefore, the cytotoxicity of the selected extracts was determined for three mammalian cell models, namely, a human hepatocellular carcinoma (HepG2) (if it was considered as a bioactive compound for oral dosage forms) and mouse melanoma (B16 4A5) (for a potential use as skin dosage forms), and one non-tumor cell line (human embryonic kidney, HEK 293). Incubating cells with the sea lavender flower extracts at a concentration of 100 µg/mL for 72 h resulted in cellular viabilities above 91% (Figure 1). 

The extracts triggered an increase in cell viability of HEK 293 and HepG2 cells (133–147% and 165–185%, respectively) (Figure 1). Former work on infusions made from this species also reported nil toxicity in mammalian cells (HepG2, mouse stromal bone marrow [S17] and mouse microglia [N9]), and against the brine shrimp *Artemia salina* [26]. Therefore, both extracts of sea lavender flowers (freshwater and 300 mM NaCl) can be considered non-toxic, suggesting that they can be used as safe ingredients for use in herbal health supplements and/or food additives. 

## 4. Conclusions

This work reported on the effect of the irrigation salinity on the chemical and functional properties of water extracts obtained from greenhouse-produced sea lavenders. The irrigation salinity and plant organ affected the in vitro antioxidant capacity and chemical composition of the extracts, which were mainly composed of flavonoids and its derivatives (e.g., sulphated, methylated, and glycosylated), but both properties were preserved under fresh- and saltwater irrigation (up to 300 mM NaCl). The flower extract had the highest chemical diversity and in vitro antioxidant properties and exhibited high ex vivo antioxidant capacity in OxHLIA and TBARS assays, inhibition of tyrosinase, and was not toxic. Our results suggest that flowers from sea lavender cultivated in greenhouse conditions and irrigated with aquaculture wastewater with a concentration up to 300 mM NaCl could provide a flavonoid-rich water extract with potential uses in herbal health supplements and/or food additives, for its antioxidant and tyrosinase inhibitory properties. 

## Figures and Tables

**Figure 1 foods-10-03104-f001:**
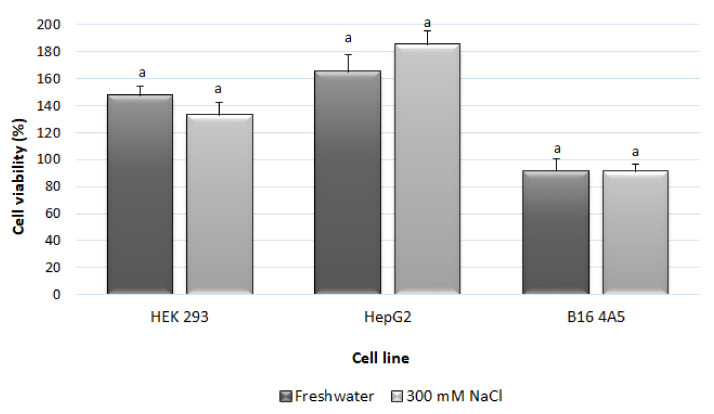
Cytotoxicity of cultivated sea lavender flower extracts irrigated with freshwater and 300 mM NaCl towards non-tumoral human embryonic kidney (HEK 293), human hepatocellular carcinoma (HepG2) and mouse melanoma (B16 4A5) cell lines. Results are expressed as cellular viability (%) at a concentration of 100 µg/mL after 72 h. Values represent the mean ± standard error of the mean (SEM) of at least six repetitions (*n* = 6). For each cell line, values followed by the same letter (a) are significantly similar at *p* < 0.05 (Tukey HSD test).

**Table 2 foods-10-03104-t002:** In vitro antioxidant activities of water extracts of different sea lavender (L. algarvense) organs (flowers, peduncles, and leaves) from greenhouse-produced plants irrigated with different salinities (freshwater, and 300 and 600 mM NaCl). Results are expressed as IC50 values (µg/mL).

Irrigation Salinity/Treatment	Plant Organ	DPPH	ABTS	CCA	FRAP
Freshwater	Flower	604 ± 4 ^d^	397 ± 2 ^b^	642 ± 21 ^b^	129 ± 6 ^a^
	Peduncles	532 ± 17 ^c^	800 ± 13 ^e^	922 ± 31 ^d^	251 ± 22 ^c^
	Leaves	549 ± 7 ^c^	793 ± 11 ^e^	953 ± 27 ^d^	339 ± 14 ^d^
300 mM NaCl	Flower	692 ± 11 ^f^	617 ± 14 ^c^	720 ± 8 ^c^	191 ± 17 ^b^
	Peduncles	383 ± 7 ^b^	745 ± 9 ^d^	-	228 ± 10 ^c^
	Leaves	-	-	-	351 ± 11 ^d^
600 mM NaCl	Leaves	-	-	-	251 ± 7 ^c^
Positive control *		111 ± 9 ^a^	142 ± 11 ^a^	171 ± 9 ^a^	na

-: activity lower than 50% at the higher concentration tested (1 mg/mL). na: not applicable. * Positive controls: BHT (RSA of DPPH and ABTS) and EDTA (CCA). Values represent the mean ± standard error of the mean (SEM) of at least three repetitions performed in triplicate (*n* = 9). In the same column, values marked with different letters are significantly different at *p* < 0.05 (Tukey HSD test).

**Table 3 foods-10-03104-t003:** Functional properties of water extracts from greenhouse-cultivated sea lavender (*L. algarvense*) irrigated with freshwater and aquaculture wastewaters containing 300 mM of NaCl: ex vivo antioxidant (OxHLIA and TBARS), in vitro anti-melanogenic/anti-browning (tyrosinase inhibition), cellular anti-inflammatory (NO reduction), and inhibition of melanin synthesis on mouse melanoma (B16 4A5) cells. Results are expressed as IC_50_ values (µg/mL).

Biological Activity	Method	Freshwater	300 mM	Positive Control *
Antioxidant	TBARS	127 ± 45 ^c^	81 ± 28 ^b^	9.1 ± 0.3 ^a^
	OxHLIA (Δt = 60 min)	136 ± 4 ^b^	140 ± 4 ^b^	21 ± 1 ^a^
Anti-inflammatory	NO reduction	-	-	16 ± 1
Anti-melanogenic/anti-browning	Tyrosinase inhibition	-	873 ± 59 ^b^	137 ± 6 ^a^
Anti-melanogenic	Inhibition of melanin synthesis by B16 4A5 cells	-	-	16 ± 1

-: activity lower than 50% at the maximum concentration tested (NO reduction: 400 µg/mL; tyrosinase inhibition: 1000 µg/mL; inhibition of melanin synthesis on B16 cells: 100 µg/mL); *: positive controls: Trolox (antioxidant), dexamethasone (anti-inflammatory), arbutin (anti-melanogenic/anti-browning) and ellipticine (cytotoxicity). TBARS: inhibition of lipid peroxidation using thiobarbituric acid reactive substances; OxHLIA: oxidative haemolysis inhibition assay; NO: nitric oxide. Values represent the mean ± standard error of the mean (SEM) of at least six repetitions (*n* = 6). In each line, values followed by different letters are significantly different at *p* < 0.05 (Tukey HSD test).

## Data Availability

The dataset is available upon request from the corresponding author.

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
