# Peer review of "Metabolomic Profile and Biological Properties of Sea Lavender (Limonium algarvense Erben) Plants Cultivated with Aquaculture Wastewaters: Implications for Its Use in Herbal Formulations and Food Additives"

_foods, 2021, doi:10.3390/foods10123104_

Round 1

Reviewer 1 Report

The study is good and informative.

I have some questions and doubts.

1. It has been mentioned that 1,2,18,20,23 compounds tend to increase with increase in salinity, Is there any specific reason for that? As all these compounds are glucose derivatives. Would that somehow be relevant to our studies or targeted activities?

2. Why there are no valued obtained at 600mM NaCl concentration for ABTS, DPPH assays?

3.  Why only extreme salt concentration taken? Why in between values like 100 or 200mM was not taken for salinity stress??

4. Why no other skin dosage form related cell line or no other non tumor cell line?

5. The total phenolic and flavanoid content could have been studied and its correlation studies could have been performed for better understanding of the results.

Examples to follow:

Metabolite profiling, antioxidant, scavenging and anti-proliferative activities of selected tropical green seaweeds reveal the nutraceutical potential of Caulerpa spp.

Antioxidant, Scavenging, Reducing, and Anti-Proliferative Activities of Selected Tropical Brown Seaweeds Confirm the Nutraceutical Potential of Spatoglossum asperum

Author Response

Dear Reviewer, 

Thank you for your comments. Below I present my answers to your questions.

Question 1:  It has been mentioned that 1,2,18,20,23 compounds tend to increase with increase in salinity, Is there any specific reason for that? As all these compounds are glucose derivatives. Would that somehow be relevant to our studies or targeted activities?

Answer: The salt stress enhances the accumulation of glucose derivatives for stress mitigation, which includes osmoprotection, carbon storage, and scavenging of reactive oxygen species. Thus, accumulation of those metabolites under stress may also affect the biological activities presented by the produced biomass. – This information was included in the discussion section.

Question 2: Why there are no valued obtained at 600mM NaCl concentration for ABTS, DPPH assays?

Answer: Table 2 presents the IC50 values for the antioxidant activity of the sea lavender extracts. The values for 600 mM NaCl were not presented because the percentage of activity was lower than 50% at the maximum concentration tested. This information is indicated in the table caption “-: activity lower than 50% at the higher concentration tested (1 mg/mL)”.

Question 3: Why only extreme salt concentration taken? Why in between values like 100 or 200mM was not taken for salinity stress??

Answer: These extreme salt concentrations were selected because they are similar to those that occur in the natural environment of this species.

Question 4: Why no other skin dosage form related cell line or no other non tumor cell line?

Answer: Additional cell models could be used; however, these cell lines were selected once they are the most relevant amongst the available ones in our research centre at the time of the experiment.

Question 5: The total phenolic and flavanoid content could have been studied and its correlation studies could have been performed for better understanding of the results.

Answer: The total phenolic and flavonoid content were not determined for this study, since a more detailed analysis of the main metabolites of water extracts of the sea lavender was made by liquid chromatography-tandem high-resolution mass spectrometry (LC-ESI-HRMS/MS), which provides the structural identification of the individual components with higher specificity and sensitivity than the total contents assays.

Reviewer 2 Report

This work presents interesting results about sea lavender (Limonium algarvense Erben). However, I believe that some points in the manuscript should be better discussed to increase the robustness of the conclusions. Mainly regarding its potential use in herbal health supplements and/or food additives.

The authors present data from a metabolic analysis performed on the LC-HRMS/MS. The results are very interesting, but they only allow for a discussion between tested treatments. Because, as this is a qualitative analysis, the comparison between these results with other species (halophyte or glycophyte) is hampered.

On the other hand, quantitative values of different bioactive activities are presented. However, I didn't understand the antioxidant capacity (against different radicals) of the plants cultivated with saline high salinity (300 and 600 mM) they are not shown in Table 2. Furthermore, the antioxidant capacities for the different organs of "sea lavender" were lowest than that of the used positive controls.

Based on the above, I believe the conclusions should be rewritten. Being more consistent with the data presented throughout the manuscript.

Author Response

Dear Reviewer, 

Thank you for your comments. Below I present my answers.

Comment 1: However, I didn't understand the antioxidant capacity (against different radicals) of the plants cultivated with saline high salinity (300 and 600 mM) they are not shown in Table 2. Furthermore, the antioxidant capacities for the different organs of "sea lavender" were lowest than that of the used positive controls.

Answer: Table 2 presents the IC50 values for the antioxidant activity of the sea lavender extracts at different assays; however, some samples (300 and 600 mM NaCl) did not reach 50% of activity, which not allowed for the calculation of IC50 values. This information is indicated in the table caption “-: activity lower than 50% at the higher concentration tested (1 mg/mL)”. Moreover, is common that raw extracts present lower activity than pure compounds used as standards, since the effective concentration of each of active compound present in the raw extract will be much lower than that used for the positive control. Also, raw extracts represent a mixture of hundreds or thousands of compounds that can interact with each other, resulting in antagonistic effects and reduced activity of the mixture.

Comment 2: Based on the above, I believe the conclusions should be rewritten. Being more consistent with the data presented throughout the manuscript.

Answer: The conclusions were modified as suggested.

Reviewer 3 Report

Manuscript is well organized and well presented however it needs some improvement.

Carefully check for technical errors such as - Line 178 CO2

Next to each method write how many repetitions were conducted. In statistical analysis is written at least 3, but be specific.

Table 2 – rearrange text – something is wrong with the text in line 337;

Table 2 and 3: -: - what is the meaning of this

Section should be named Results and discussion not just results

Carefully check and correct references and write them according to author guidelines

Author Response

Dear reviewer, 

Thank you for your comments. Below I present my answers.

Comment 1: Carefully check for technical errors such as - Line 178 CO2

Answer: The error was corrected.

Comment 2: Next to each method write how many repetitions were conducted. In statistical analysis is written at least 3, but be specific.

Answer: The number of repetitions is indicated in the respective caption of each table and figure.

Comment 3: Table 2 – rearrange text – something is wrong with the text in line 337;

Answer: Text was rearranged.

Comment 4: Table 2 and 3: -: - what is the meaning of this

Answer: The meaning of the symbol “-“ is indicated in the table caption “-: activity lower than 50% at the higher concentration tested”.

Comment 5: Section should be named Results and discussion not just results

Answer: The name of the section was corrected to Results and discussion.

Comment 6: Carefully check and correct references and write them according to author guidelines

Answer: The reference list was revised according to the journal guidelines.